# Standardized Classification of Cerebral Vasospasm after Subarachnoid Hemorrhage by Digital Subtraction Angiography

**DOI:** 10.3390/jcm11072011

**Published:** 2022-04-03

**Authors:** Helena Merkel, Dirk Lindner, Khaled Gaber, Svitlana Ziganshyna, Jennifer Jentzsch, Simone Mucha, Thilo Gerhards, Sabine Sari, Annika Stock, Felicitas Vothel, Lea Falter, Ulf Quäschling, Karl-Titus Hoffmann, Jürgen Meixensberger, Dirk Halama, Cindy Richter

**Affiliations:** 1Department of Neuroradiology, Leipzig University Hospital, Liebigstraße 20, 04103 Leipzig, Germany; helena.merkel@medizin.uni-leipzig.de (H.M.); jennifer.jentzsch@medizin.uni-leipzig.de (J.J.); simone.mucha@medizin.uni-leipzig.de (S.M.); thilo.gerhards@medizin.uni-leipzig.de (T.G.); lea.falter@medizin.uni-leipzig.de (L.F.); ulf.quaeschling@ksbl.de (U.Q.); karl-titus.hoffmann@medizin.uni-leipzig.de (K.-T.H.); 2Department of Neurosurgery, Leipzig University Hospital, Liebigstraße 20, 04103 Leipzig, Germany; dirk.lindner@medizin.uni-leipzig.de (D.L.); khaled.gaber@medizin.uni-leipzig.de (K.G.); juergen.meixensberger@medizin.uni-leipzig.de (J.M.); 3Department of Anaesthesiology, Leipzig University Hospital, Liebigstraße 20, 04103 Leipzig, Germany; svitlana.ziganshyna@medizin.uni-leipzig.de; 4Department of Neuroradiology, Giessen University Hospital, Klinikstraße 33, 35392 Giessen, Germany; sabine.sari@uk-gm.de; 5Department of Neuroradiology, Würzburg University Hospital, Josef-Schneider-Straße 2, 97080 Würzburg, Germany; stock_a@ukw.de; 6Department of Radiology and Neuroradiology, Sana Hospital Borna, Rudolf-Virchow-Straße 2, 04552 Borna, Germany; felicitas.vothel@sana.de; 7Department of Oral and Maxillofacial Surgery, Leipzig University Hospital, Liebigstraße 12, 04103 Leipzig, Germany; dirk.halama@medizin.uni-leipzig.de

**Keywords:** cerebral vasospasm, subarachnoid hemorrhage, classification, vessel diameter

## Abstract

Background: During the last decade, cerebral vasospasm after aneurysmal subarachnoid hemorrhage (SAH) was a current research focus without a standardized classification in digital subtraction angiography (DSA). This study was performed to investigate a device-independent visual cerebral vasospasm classification for endovascular treatment. Methods: The analyses are DSA based rather than multimodal. Ten defined points of intracranial arteries were measured in 45 patients suffering from cerebral vasospasm after SAH at three time points (hospitalization, before spasmolysis, control after six months). Mathematical clustering of vessel diameters was performed to generate four objective grades for comparison. Six interventional neuroradiologists in two groups scored 237 DSAs after a new visual classification (grade 0–3) developed on a segmental pattern of vessel contraction. For the second group, a threshold-based criterion was amended. Results: The raters had a reproducibility of 68.4% in the first group and 75.2% in the second group. The complementary threshold-based criterion increased the reproducibility by about 6.8%, while the rating deviated more from the mathematical clustering in all grades. Conclusions: The proposed visual classification scheme of cerebral vasospasm is suitable as a standard grading procedure for endovascular treatment. There is no advantage of a threshold-based criterion that compensates for the effort involved. Automated vessel analysis is superior to compare inter-group results in research settings.

## 1. Introduction

The diagnosis of cerebral vasospasm following subarachnoid hemorrhage (SAH) is still challenging without any standardized classification. Digital subtraction angiography (DSA) is considered the gold standard to assess the severity of cerebral vasospasm and its impact on subsequent perfusion [1,2,3]. The angiographic manifestation of cerebral vasospasm has not yet been systematically evaluated. It is unknown whether unique characteristics such as distribution pattern, number of affected arteries, the degree of severity, or the treatment response allow for conclusions regarding the further clinical course and patients’ outcome.

Literature research reveals many incomparable classification schemes of cerebral vasospasm. The most commonly used criterion for diagnosing severe vasospasm was a reduction in vessel caliber >50%, moderate if between 25–50%, and mild if <25% of different intracranial vessels [4,5]. Afat et al. [6] and Neulen et al. [6] graded different vessel segments according to the following system: 0, no vasospasm; 1, vasospasm with <50% change in the vessel diameter; 2, vasospasm with 50% narrowing compared to the initial DSA. Kerz et al. [7] scored the severity of angiographic vasospasm before and after vasodilator infusion with three grades: Grade 1 = up to 30% reduction in the arterial diameter compared to the diameter in the diagnostic angiography (pre-spasm), grade 2 = 30–70%, reduction and grade 3 > 70% reduction. Weidauer et al. [8] differentiated between proximal and distal cerebral vasospasm. Diameters of the proximal segments of the middle cerebral artery (MCA), anterior cerebral artery (ACA), the posterior cerebral artery (PCA), the distal part of the internal carotid artery (ICA), as well as the intradural vertebral artery (V4 segment) and the basilar artery were measured in absolute values. These values were proportioned to the absolute values of the extradural petrous segment of the ICA or the extradural vertebral artery (V3 segment). The presence of cerebral vasospasm was classified as none (0% to 10%), mild (11% to 33%), moderate (34% to 66%), or severe (67% to 100%) vascular narrowing. Focal (<50% of the segment length) or diffuse vasospasm (>50% of the segment length) were distinguished. None of these classification schemes underwent validation.

In all previous studies, the measured points of the graded vessels were not precisely defined. Most publications assessed the value of diameter changes in ICA, MCA, ACA, and PCA segments as equal. Although, a recent retrospective study of Ditz et al. [9] of 85 patients found angiographic cerebral vasospasm predominantly involving the anterior circulation in concordance with previous reports [10]. In most cases, 2–3 vessels were affected [9].

The diagnosis was usually made by comparing the DSA on admission with the DSA at the time of suspected cerebral vasospasm. Early angiographic vasospasm on admission was not taken into account [11,12,13]. To our knowledge, reference values for intracranial arteries on DSA are still not systematically defined.

The goal of this study was to develop an easy-to-use standardized classification of cerebral vasospasm after SAH. Defined reference points of cerebral vessel segments of patients suffering from vasospasm served to validate a visual segment-based classification with the aid of mathematical clustering.

## 2. Materials and Methods

### 2.1. Study Design and Recruitment

All consecutive patients presenting to our hospital from January 2014 to July 2021 with acute aneurysmal SAH and subsequent cerebral vasospasm who had undergone at least one intra-arterial spasmolysis were retrospectively recovered from our radiology information system.

The local Ethics Committee of the Medical Faculty approved this study (reference number: 412/19-ek), and all participants gave their informed consent. Inclusion criteria were: (1) an aneurysmatic SAH verified by CT, MRI, or lumbar puncture, and (2) pharmacological intra-arterial spasmolysis. Patients with mycotic or traumatic-induced pseudoaneurysms, aneurysms associated with arteriovenous malformations or dissections were excluded. Parameters that influenced the clinical outcome independent from cerebral vasospasm as sepsis (*n* = 1), chronic ICA occlusion (*n* = 1), and recurring seizures (*n* = 1) were excluded as well.

The retrieved clinical information included demographic data (age, sex) and the initial SAH-related patient characteristics (Hunt and Hess scale, further aneurysms, type, and time-point of treatment of the vascular lesion). One author retrospectively graded each admission non-contrast head CT for SAH using the Fisher scale. Additionally, all non-contrast CTs were examined to define hemorrhagic and ischemic infarctions before and after aneurysm treatment and vasospasm-related infarctions. Delayed cerebral ischemia (DCI) was defined as the occurrence of vasospasm-related infarction. The functional outcome was routinely assessed after six months in clinical reports. The modified Ranking Scale (mRS) was retrospectively assessed by one author.

### 2.2. Standard SAH Treatment Protocol

All patients were monitored in our neurointensive care unit. Nimodipine was given from the day at admission orally (6 × 60 mg/day) or via gastric tube. Mean arterial blood pressure was always sustained above 80 mmHg to avoid reduced cerebral perfusion. A local standard operating procedure for SAH was used (not included).

### 2.3. CVS Monitoring

Monitoring included daily transcranial Doppler investigation (TCD) of cerebral arteries. Cerebral vasospasm was suspected based on altered level of consciousness or new neurological impairment (*n* = 16) or increased flow velocity in TCD above 160 cm/s in MCA or doubling within 24 h (*n* = 20). CT was performed in 44 patients to detect new territorial infarctions (*n* = 5). CT perfusion analysis was additionally performed in cases with ambiguous findings (*n* = 29). Six patients revealed a perfusion deficit leading to intra-arterial spasmolysis. Diagnostic DSA was performed if clinical worsening persisted, but TCD and CT-perfusion did not reveal cerebral vasospasm (*n* = 1).

### 2.4. Diagnostic Cerebral Angiography

Diagnostic DSA was performed using a biplane system (Axiom Artis, Siemens, Erlangen, Germany or AlluraClarity, Philips Healthcare, Best, The Netherlands) or a monoplane system (Innova 4100; GE Healthcare, Waukesha, WI, USA). Iopromid (60–120 mL, containing 300 mg iodine per mL) was used as the contrast agent.

Every patient who underwent intra-arterial spasmolysis obtained a diagnostic DSA run to assess the severity of angiographic cerebral vasospasm. DSA was performed according to vascular regions with suspected cerebral vasospasm. A visual comparison of the diagnostic DSA and DSA on admission, if applicable, was performed. Therefore, not all segments were necessarily examined in every patient at every time point, especially not the vertebrobasilar circulation.

### 2.5. Early Angiographic Vasospasm

The onset of early angiographic vasospasm was evaluated according to Jabbarli et al. [13]. Angiograms were analyzed up to 72 h after SAH for pre-existing early angiographic vasospasm and scored with the BEHAVIOR score according to Jabbarli et al. [13]. Early angiographic vasospasm was evaluated on diagnostic angiograms before intracranial aneurysm treatment without spasmolytic therapy. One patient sustained SAH during elective endovascular treatment being excluded from group SAH 0. The BEHAVIOR score can be divided into three levels: low (0 to 2 points, *n* = 2), medium (3 to 6 points, *n* = 18), and high risk (7 to 11, *n* = 6) of DCI development.

### 2.6. Intra-Arterial Spasmolysis

We practiced intra-arterial administration of spasmolytics over the entire period regularly, first using Nimodipine, later Milrinone, or a combination of both. Nitro-glycerine and Alprostadil were added as expanded access if the effect of primarily given substances was insufficient. Each pharmacological agent shows varying effects on vessel segments (results are not shown), requiring further investigations in ongoing studies.

### 2.7. Measured Points on Angiograms

In lateral projections, diameters of the extradural internal carotid artery (ICA: C5) and intradural internal carotid artery (ICA: C6) were measured. The remaining segments of the intradural internal carotid artery (ICA: C7, narrowest point of C7 = nC7), the middle cerebral artery (MCA: proximal M1 = pM1, distal M1 = dM1, M2), and the anterior cerebral artery (ACA: proximal A1 = pA1, distal A1 = dA1, A2) were measured in posterior-anterior projections. The measured points and anatomical variations were double-checked by two readers.

All values were measured on a diagnostic workstation (syngo.plaza, VB30C, Siemens Healthcare, Erlangen, Germany) according to a defined protocol:

Lateral projection:C5: 2 mm proximal to the origin of the ophthalmic artery.C6: at the origin of the ophthalmic artery.

Posterior-anterior projection:C7: 2 mm proximal to the carotid T.nC7: the narrowest location in vasospastic arteries of C7 segments.pM1: 2 mm distal to carotid T.dM1: 2 mm proximal to middle cerebral artery bifurcation.M2: 4 mm distal to the M1/M2 transition.pA1: 2 mm distal to carotid T.dA1: 2 mm proximal to change of course.A2: 2 mm distal to the anterior communicating artery.

The rater decided whether vessel narrowing was presumably due to a spasm or a pre-existing condition, such as developmental hypoplasia or atherosclerosis. The proximal ACA was considered hypoplastic if the contralateral A1-segment and the AComA were large and the A2-segments were well filled. Confidence in identifying A1 hypoplasia was increased if no spasm was present in the ipsilateral A2 segment. In the case of the proximal bifurcation variant of the MCA, the most prominent branch of the M1 segment was evaluated. The most prominent M2 branch was measured. If there was a trifurcation or quattrofurcation, M2 diameters were predictable smaller.

The values were not taken in material-overlapping areas due to coiling or vessel widening due to aneurysms at measure points, respectively. Vessel segments with stents of any kind were excluded.

### 2.8. Visual Cerebral Vasospasm Classification

In concordance with previous reports [9,10], we assumed a pattern of cerebral vasospasm spreading from distal to proximal segments, first affecting the ACA (Figure 1). Therefore, four cerebral vasospasm grades (CVSG) from 0 to 3 are defined in Table 1.

The rare appearance of vertebrobasilar ischemia after SAH made an implication of the vertebrobasilar vasospasm in the classification scheme not maintainable. Furthermore, angiographic runs of the vertebrobasilar system matter for radiation exposure and entrail additional risks of interventional or circulatory complications.

Altogether, according to this visual cerebral vasospasm classification, 237 DSA examinations at three time points (hospitalization, prespasmolytic, and control after six months) were graded by six raters. The raters consisted of six neuroradiologists with a minimum of six months of angiographic experience divided into two groups of two and four neuroradiologists, respectively. The DSAs were pseudonymized and arranged in PowerPoint slides. The first slides offered the cerebral vasospasm classification in Table 1 with drawings and angiograms in Figure 1. There was no initial training performed for the raters. The DSA on admission and/or the follow-up DSA after six months were shown to assess anatomical variants first for each case. Second, the prespasmolytic DSAs were presented chronologically. It was pointed out to the raters that DSAs on admission can display early angiographic vasospasm or manifest vasospasm due to a recent SAH with an unknown time-point. The 6-month-follow-up DSA was used as a reference when evaluating vasospasm. Hypoplastic angiograms due to large territorial infarction were excluded from the rating.

The neuroradiologists of the first group rated only angiograms to the defined criteria (Table 1). For the second group, the diameter of the proximal M1 segment was quoted on each slide as a complementary criterion. The pM1 measured point less or equal to 1 mm served as the threshold for CVSG 3.

### 2.9. Statistical Analysis

Statistical analyses were performed with SPSS version 27.0 (IBM Corporation; New York, NY, USA) and R version 4.0.1 (The R Foundation for Statistical Computing, Indianapolis, IN, USA). Data were analyzed using chi-square-test and clustering as appropriate. A 2-tailed value of *p* < 0.05 indicated statistical significance.

### 2.10. Clustering of CVS Values

Data grouping (or data clustering) is a method that can form classes of objects with similar characteristics. This method partitions the dataset of vessel diameters into clusters. According to defined criteria, the diameters in the same cluster are more similar than diameters in different clusters.

First, data type differences are analyzed by calculating the dissimilarity matrix. It is a mathematical expression of how different the points in the data set are from each other to form groups. According to previous data set analysis, three or four clustering groups (*k*-means) were the most suitable to include as many values as possible and exclude as few values as necessary. Corresponding to the four grades of visual classification, the *k*-means algorithm of four was predefined. The clustering was repeated 25 times to get as precise as possible. The four groups of clustering were recoded in the four grades of visual cerebral vasospasm classification according to their distribution of vessel values. In the following, clustered CVSG is termed cCVSG.

## 3. Results

### 3.1. Study Group

Forty-five patients experienced protracted cerebral vasospasm after SAH (Table 2). These patients were aged 29 to 79 (mean: 49.6) and predominantly female (2:1). Patients hospitalized on the day of SAH (SAH 0) had no significantly higher Fisher score (*p* = 0.106) and Hunt and Hess (HH, *p* = 0.151) compared to the remaining patients. All patients underwent endovascular treatment or clipping (coiling *n* = 33, clipping *n* = 5, coiling and clipping *n* = 3, flow diversion *n* = 2, coiling and flow diversion *n* = 2) within 48 h after SAH when immediately admitted to hospital or at time of hospitalization. One patient with flow diversion was excluded due to vessel diameter manipulation, at least at the implant site. The remaining two patients with flow diversion revealed significant vasospasm on the contralateral side analyzed in this study. In one case, no hemorrhage source was found. Patients with craniectomy, as for aneurysm clipping, were excluded.

Intracranial pressure was compensated to <20 mmHg via external ventricular drainage in 35 cases on the day of hospitalization before the first DSA. Ten patients did not show clinical signs of increased intracranial pressure or disturbed cerebrospinal fluid drainage being treated without external ventricular drainage. Most patients (*n* = 24) treated on SAH day already possessed external ventricular drainage, whereas only three patients did not.

Vessel diameters in DSA were taken at the following three-time points (number of analyzed angiograms):Hospitalization due to SAH (*n* = 34).Before intra-arterial spasmolysis (*n* = 183).Follow-up examination after six months (*n* = 20).

Six patients died, and twelve patients did not undergo control DSA due to poor clinical outcomes (mRS 4–5). Clipping did not require follow-up DSA after 6 months (mRS 1 *n* = 1, mRS 5 *n* = 2, mRS 6 *n* = 2). Four patients with poor outcomes due to large territorial infarcts with consecutive hypoplasia of the providing arteries were excluded. Two patients did not yet receive their follow-up examination (mRS 4 at discharge).

### 3.2. Clustering

The vessel diameters of the ten measured points were underwent mathematical clustering with the *k*-means algorithm of four and matched with the four grades of CVSG (0–3), named cCVSG. The distribution of vessel diameters after mathematical clustering is represented in Figure 2. All vessel segments, except C5 (*p* = 0.013), had a highly significant difference of CVSG 0 to 3 (*p* < 0.001). Measurements of C5 revealed only a low significant intergrade difference of CVSG 0 to CVSG 3 (*p* < 0.05). The ACA and M2 segments presented the main difference between cCVSG 0 and 1. The proximal and distal M1 segment constricted progressing from cCVSG 0 to cCVSG 3. The intradural internal carotid artery segments show a sustained decrease in vessel diameter, with the main peak between cCVSG 2 and cCVSG 3. The extradural segment of the ICA, C5, is only affected in cCVSG 3.

### 3.3. Rater-Agreement and Reproducibility

The clustering distribution according to the grades 0–3 is shown in Figure 3 and Table 3. The results were re-evaluated after the grading by two interventional neuroradiologists (group 1: NR 1 and NR 2) with an agreement of 68.4%. There was no significant intra-group difference (*p* = 0.206), but the second rater had difficulties distinguishing CVSG 1 and 3. The highest agreement with the mathematical clustering was reached in CVSG 0 and 2.

For the second group, the classification was modified with an additional criterion to separate CVSG 2 from CVSG 3. The pM1 measured point less or equal to 1 mm with a 95% confidence interval was chosen as the threshold for CVSG 3. The measured values for pM1 were added on each slide. Four neuroradiologists (group 2: NR 3–6) used the modified classification. The results showed an increased reproducibility of 75.2% with 100% agreement of four raters. There were no significant intra-group differences (*p* = 0.310).

CVSG 1 ratings decreased by half, whereas CVSG 2 ratings were significantly more distinct. The additional information of one vessel diameter remarkably influenced the raters of the second group in their decision. There was a highly significant inter-group difference (*p* < 0.001). The results of the second group deviated more from the mathematical clustering in cCVSG 0 and 1. For the overall grades, the pM1 value augmented the inter-rater reproducibility, but it did not remarkably influence the agreement for CVSG 3 ratings in group 2.

### 3.4. Early Angiographic Vasospasm

Twenty-seven patients received a diagnostic angiography before aneurysm treatment up to 72 h after SAH. On day 0, twelve of 18 patients were classified with cCVSG 0, five with cCVSG 1, and one with cCVSG 2. On day 1 (*n* = 4), day 2 (*n* = 4) and day 3 (*n* = 1), the examined patients did not show any vasospasm.

Four of six patients (66.7%) suffering from early angiographic vasospasm showed an unfavorable outcome with mRS 4–6. Three patients of this group died. In contrast, 10 of 21 patients (47.6%) without early angiographic vasospasm revealed an unfavorable outcome with two deaths. There was no significant correlation between early angiographic vasospasm and DCI occurrence for the 18 patients examined on day 0 (*p* = 0.429). The early angiographic vasospasm (*p* = 1), the BEHAVIOR Score (*p* = 0.573) and the risk level of BEHAVIOR Score (*p* = 0.618) for day 0–3 did not significantly predict DCI.

### 3.5. Vertebrobasilar Vasospasm

In 36 cases, the vertebrobasilar vessels were depicted angiographically. Patients suffering from vertebrobasilar vasospasm (*n* = 8) reached more often a handicapped (mRS 3–4: 2) or unfavorable clinical result (mRS 5–6: 4). The small number of cases could not show a significant mathematical influence on the outcome (*p* = 1.0) or DCI (*p* = 0.903).

### 3.6. Overall Outcome

Patients suffering from DCI (*n* = 35) revealed the following outcome after six months: mRS 0 (*n* = 2), mRS 1 (*n* = 7), mRS 2 (*n* = 2), mRS 3 (*n* = 3), mRS 4 (*n* = 8), mRS 5 (*n* = 7) and mRS 6 (*n* = 6). Patients without DCI displayed an outcome of mRS 1 (*n* = 5), mRS 3 (*n* = 1) and mRS 5 (*n* = 4).

Each decedent suffered from DCI, but an unfavorable outcome of mRS 4 to 6 was not always related to DCI. The six patients died on days 9, 14, 15, 16, 16 and 18 after SAH occurrence. Four patients with mRS 5 already suffered from an intracerebral hemorrhage on admission, underwent neurosurgical treatment and were excluded from outcome analysis. Systemic inflammatory response syndrome (*n* = 1) and sepsis (*n* = 5), pneumonia (*n* = 6) and renal failure (*n* = 4) worsened the clinal course, too.

The cCVSG on admission (*p* = 0.597), the cCVSG at the first spasmolysis (*p* = 0.123) and the highest cCVSG (*p* = 0.637) did not predict DCI. There was no significant predictability of the outcome in mRS from cCVSG on admission (*p* = 0.433), cCVSG on first spasmolysis (*p* = 0.174) and highest cCVSG (*p* = 0.572).

## 4. Discussion

Although cerebral vasospasm following aneurysmal SAH has been described over many decades [14], no standardized angiographic classification allows inter-group comparison of research findings. We investigated the characteristic angiographic pattern of cerebral vasospasm by visual classification accompanied by a mathematical clustering algorithm at ten evaluated reference points. Clustering is one of the most common unsupervised machine learning tasks [15] allowing objective classification of vessel diameters without predefined thresholds. The distribution of mathematically clustered vessel diameters matched the criteria of the visual classification in many points. The ACA segments and the M2 segment showed their main difference of diameters between cCVSG 0 and cCVSG 1. Visually detectable cerebral vasospasm was suspected to begin in these segments. The segments of the intradural ICA showed the main difference of diameters between cCVSG 2 and cCVSG 3, being the primary criterion for severe vasospasm. Severe vasospasm also affects the extradural C5 segment of the ICA.

The visual cerebral vasospasm classification was reproducible in 68.4% (group 1) and the inter-rater agreement increased to 75.2% (group 2) when threshold-based complementary criteria were added. There was no significant intra-group difference. The measuring error is not included because the pM1 values were preset. The pM1 segment is scarcely affected by anatomical variants, nearly never overprojected, an unusual aneurysm location, and easy to identify. Smaller vessel segments, f.e. A1, A2, and M2, are more difficult to measure because of possible overprojection or anatomical variation. Vessel segments with two or more branches were considered inappropriate for reliable assessments. On the other hand, the specified-in-advance pM1 diameters significantly changed the overall grading results. Raters reported that the pM1 value influenced the rating of remaining grades, too. They were more likely to rate CVSG 2 than CVSG 1 (Figure 3, Table 3). These data suggest that measured values have a preponderance compared to the subjective evaluation of the angiogram. They never serve as decision support alone. Changes in pM1 diameters led to an overestimation of mild cerebral vasospasm. The visual classification scheme includes all proximal intradural vessels of the anterior circulation. Isolated changes of one segment are set in relation to the overall pattern without given values. The mathematical clustering takes account of all vessel diameters in the same manner. All raters of group 2 gave more weight to the pM1 values than the overall pattern of the angiogram. In severe vasospasm, narrowing of the intradural ICA is pivotal. The C7 diameter may better serve as a complementary criterion to define CVSG 3. Similarly, slight changes of C7 diameter may even lead to an overestimation of mild vasospasm not visually detectable.

There are many factors known to play an essential role in clinical outcomes [16]. Early angiographic and vertebrobasilar vasospasms were proven as contributing factors being reported as associated with poor outcomes [12,17]. There is an option that early angiographic vasospasm is an immediate reflex of the vessel wall to the acute bleeding giving an idea of the severity of symptomatic vasospasm. Jabbarli et al. [13] invented the BEHAVIOR score to assess the risk of developing DCI. They defined groups of low, medium, and high-risk levels for the occurrence of DCI (Table 2). In our study, neither the early angiographic vasospasm alone nor the group of BEHAVIOR score correlated significantly with DCI. Jabbarli et al. used the multivariate logistic regression analysis to define DCI predictors. This statistical method requires huge data sets being inapplicable in our study.

The vertebrobasilar system was only monitored in 36 of 45 of our patients because vasospasm affects the anterior circulation more often. Ischemic lesions in the vertebrobasilar system are rare. Ubiquitary vasospasm is suggested to present with vertebrobasilar vasospasm. In our cohort, only 22.2% (*n* = 8) of patients suffered from vertebrobasilar vasospasm, even under the reported incidence of 37.8% [17]. Due to a small number of cases, there was no significant correlation with the highest cCVSG, the occurrence of DCI, or the patient outcomes.

Five patients in particular with severe intracranial hemorrhage on admission revealed an unfavorable outcome with mRS 5. Typical SAH-related complications such as pneumonia and sepsis were detected as critical determinants of outcome [16] being present also in our vasospasm cohort. DCI has emerged as the most clinically relevant sequel given the strong association with poor clinical outcomes, including cognitive impairment and quality-of-life metrics [18]. DCI occurs in about 30% of cerebral vasospasm patients, with a peak between days 3 and 14 after the initial SAH [19]. However, angiographic vasospasm does not necessarily lead to DCI. In acute stroke, poor collateral flow is associated with a worse outcome and faster growth of larger infarcts. In cerebral vasospasm, acute vessel narrowing causes acute hypoperfusion in brain parenchyma. Blood supply to the brain is secured by an extensive but individualized collateral circulation system. The extend of collateral flow is highly variable between individuals. As a consequence, the resulting infarcts are not solely dependent on the severity of vasospasm. DCI is more complex than simply being a result of arterial narrowing. A multifactorial process involving blood-brain-barrier disruption, microthrombosis, cortical spreading depolarization, and loss of cerebral autoregulation leads to DCI. In the past decades, evidence has shown that cerebral vasospasm and DCI should be considered as two separate entities. The heterogeneity in the definition of DCI has allowed for many variations of its diagnostic and therapeutic regimen [20]. These contributing factors explain the missing predictability of our visual classification for patient outcomes.

Much work has been done to elucidate factors that are predictive of cerebral vasospasm. For those patients deemed to be at high risk, therapies could be instituted earlier and more aggressively. Those patients who are thought to be at low risk could avoid the significant potential complications and costs associated with vasospasm treatment. The most valuable predictors can be easily and consistently obtained after the patient presents for treatment. DSA is the gold standard to evaluate cerebral vasospasm. In clinical settings, the decision towards therapeutic consequences as spasmolytic therapies is made according to clinical findings and non-invasive imaging such as CT, CTA and CT-perfusion. Still, there is also limited evidence regarding its reliability. Letourneau-Guillon et al. [21] assessed the inter- and intraobserver reliability of CTA in diagnosing cerebral vasospasm in the literature. In the analyzed 14 studies, eight different classification schemes were used with three to five categories and various arbitrary cutoffs. Most reports were primarily diagnostic accuracy studies, dedicated to a comparison with DSA as the gold standard, while agreement studies were limited to 2–3 raters with 22–34 patients. The interobserver agreement ranged from 66% to 79%. In their approach, Letourneau-Guillon et al. [21] asked eleven clinicians to grade the degree of vasospasm of 17 arterial segments on a 4-category scale with the initial CTA. There were three additional questions (yes/no): (1) Is there moderate-severe vasospasm at any location? (2) Presuming the presence of a new neurologic deficit corresponding to the territory of the artery affected by vasospasm, would you recommend a change in medical management? (3) Would you recommend DSA with or without balloon angioplasty? The latter two clinical decisions were based on the readers’ clinical experience. The interobserver agreement for detection of moderate-to-severe vasospasm was only fair with 34% and improved to moderate with 43% between senior raters. A perfect agreement was only found for 6 of 50 patients. The interobserver agreement on recommending DSA with or without balloon angioplasty based on CTA alone was only fair. There were only two cases in which all raters agreed that DSA should be performed.

Many well-known classification schemes, such as the Fisher scale, have been under criticism of being subjective and lacking quantitativeness [22], too. Our visual classification is device-independent, universally applicable, and does not require additional examinations. Time-consuming data exports for automated vessel analysis are not required. Despite the efforts to standardize the classification of cerebral vasospasm, the proposed grading system first seems to be complex. Simplified, it is a scheme of segment-based cerebral vasospasm manifestation starting at the ACA segments and terminating at the intradural ICA. 

TCD can recognize cerebral vasospasm in its earlier stages before it becomes clinically apparent. Unfortunately, it is not applicable for each patient if the ultrasound window is missing. The ACA and ICA are hard to measure, especially in severe vasospasm. Moreover, TCD is examiner dependent [23]. The DSA based classification can be implemented in the clinical workflow of endovascular treatment of cerebral vasospasm to develop and justify, respectively, an escalation scheme for endovascular treatment beyond a standardized evaluation of treatment success or nonsuccess. In low-grade vasospasm (CVSG 1), spasmolytic agents targeting smaller vessel segments, especially the anterior cerebral artery, could be administered preferentially. In contrast, mechanical spasmolysis for the M1 segment and the intradural ICA is more successful in moderate or severe vasospasm (CVSG 2 and 3). Pharmacological agents can be evaluated according to their vasodilating potential for different vessel segments. A moderate reproducibility of 68.4% is an argument against the new grading system, but further non-automated classifications could not show higher agreements without initial training [21,22]. In this context, we offer a scheme of visual cerebral vasospasm classification for further development.

### Limitations

This retrospective study with a focus on imaging findings has several limitations. First of all, the small, inhomogeneous sample size of the vasospasm group does not provide a robust statistical assessment even if the applied chi-square-test is robust for small sample size and all scales of measurements. Another limitation is the retrospective nature of this study. CT scans and DSA runs were only performed when clinically indicated. This might have led to a detection bias favoring patients with poor clinical outcomes. The early angiographic and the vertebrobasilar vasospasms could only be assessed for a small group of patients.

To facilitate the participation of neuroradiologists in different centers, the pM1 value was already predefined for all DSAs, not allowing any analysis for measurement errors. However, we consider the visual cerebral vasospasm classification without threshold-based criteria to be more valuable.

## 5. Conclusions

We introduce a DSA-based visual cerebral vasospasm classification for neuroradiologists in daily routine and research settings. A threshold-based complementary criterion increases the reproducibility and leads to an overestimation of cerebral vasospasm, not compensating for the effort involved. This classification needs external validation in other SAH populations to be conclusively evaluated.

## Figures and Tables

**Figure 1 jcm-11-02011-f001:**
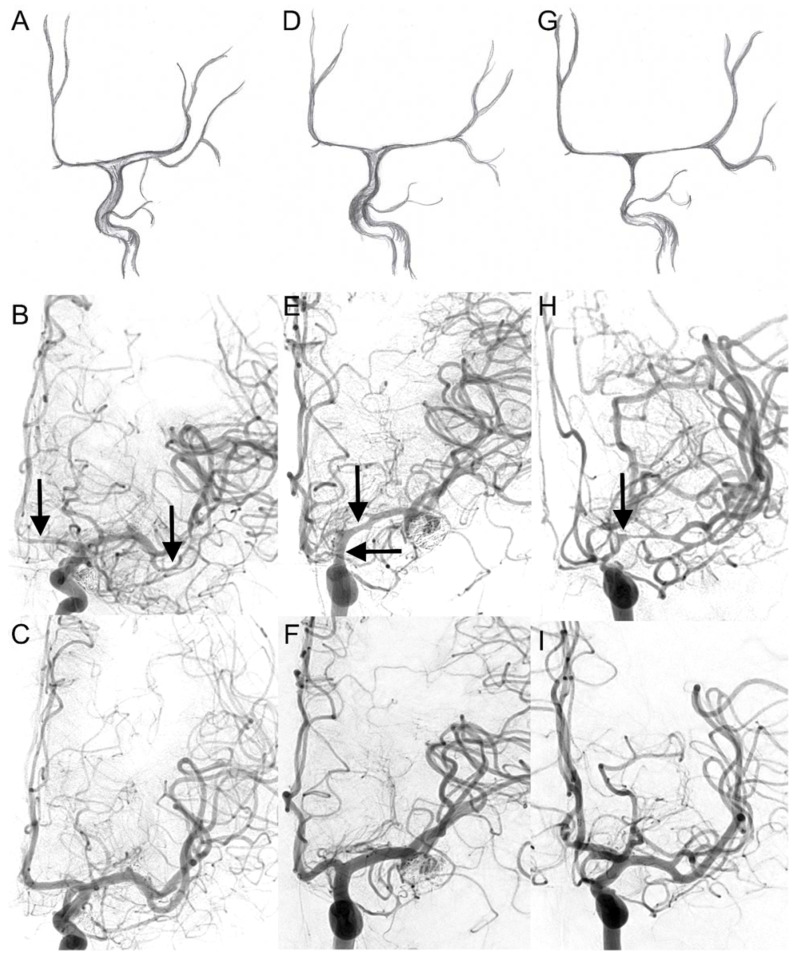
Visual cerebral vasospasm classification in digital subtraction angiography. The upper two lines display cerebral vasospasm grades (CVSG) as drawings with the corresponding angiograms. In the bottom line, the corresponding control DSA after six months is shown: (**A**,**B**) CVSG 1—Narrowing of the A2, A1, and M2 segments (arrows) with postspastic enlargement of distal M2 branches. (**D**,**E**) CVSG 2—The spasm affects the proximal M1 segment and the intradural carotid artery (arrows). (**G**,**H**) CVSG 3—The intradural carotid artery, the proximal middle cerebral artery, and the anterior cerebral artery show high-grade narrowing with a fading appearance like a ghost (arrow). (**C**,**F**,**I**)—control DSA after 6 months (CVSG 0).

**Figure 2 jcm-11-02011-f002:**
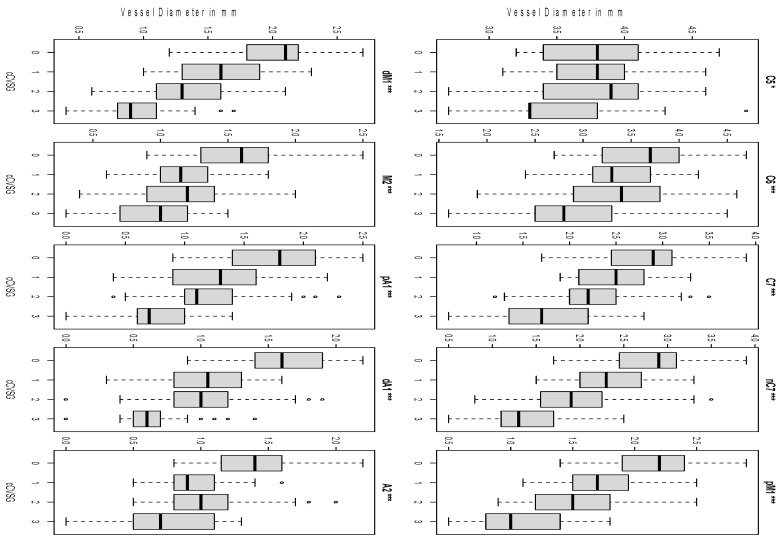
Clustering of vessel diameters. The distribution of vessel diameters according to cCVSG 0–3 after mathematical clustering is shown. All vessel segments had a significant intergrade difference (* C5 *p* = 0.013, *** the remaining *p* < 0.001). Discordant values are shown as small circles C5 clinoid segment of the internal carotid artery, C6 ophthalmic segment of the internal carotid artery, C7 terminal segment of the internal carotid artery, nC7 narrowest point of C7, pM1 proximal horizontal segment of the middle cerebral artery, dM1 distal horizontal segment of the middle cerebral artery, M2 insular segment of the middle cerebral artery, pA1 proximal pre-communication segment of the anterior cerebral artery, dA1 distal pre-communication segment of the anterior cerebral artery, A2 post-communicating segment of the anterior cerebral artery. The small circles show the discordant values.

**Figure 3 jcm-11-02011-f003:**
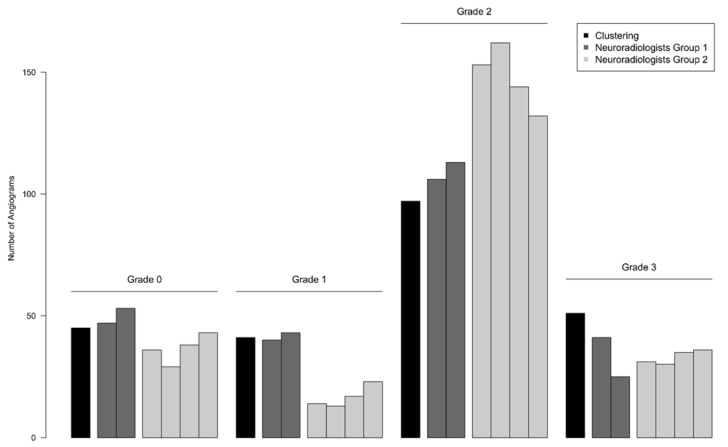
Reproducibility of visual classification of cerebral vasospasm. The mathematical clustering for objective comparison is depicted in black first, followed by the subjective grading of each rater labeled according to his profession. The first group in dark gray (two neuroradiologists) applied the primary visual cerebral vasospasm classification. The second group in light gray (four neuroradiologists) applied the modified classification supplemented with the pM1 values. Neuroradiologists of the second group deviate more from the clustering.

**Table 1 jcm-11-02011-t001:** Visual classification: Cerebral vasospasm grades (CVSG).

**Grade 0**	All intracranial vessels show a physiological shape
**Grade 1**	Vasospasm affects the A2, A1, and M2 segments
**Grade 2**	Vasospasm expands to the M1 and terminal segment of the internal carotid artery
**Grade 3**	Severe reduction in the intradural internal carotid artery with filiform A1 and M1 segments, which sometimes appears like a ghost (ghost sign)

**Table 2 jcm-11-02011-t002:** Baseline demographics.

**Clinical Characteristics**	
Patients	45
Female (%)	31 (68.9)
Mean age (range)	49.6 (29–79)
Fisher score 2–3	14
Fisher score 4	31
Hunt and Hess score 1–3	29
Hunt and Hess score 4–5	16
**Treatment**	
Timing of aneurysm treatment (in days, median, range)	0.00 (0–9)
Clipping	5
Endovascular treatment	37
Combined treatment	3
**Time-points of digital subtraction angiographies**	
On admission	34
Before intra-arterial spasmolysis	183
After 6 months (follow-up)	20
**Reasons to perform intra-arterial spasmolysis**	
Altered level of consciousness or new neurologic deficit (%)	16 (35.6)
New infarct on CT	
Perfusion impairment	3 (6.7)
Infarct before spasmolytic therapy	2 (4.4)
Both	3 (6.7)
TCD velocity increase	20 (44.4)
None other than rule out vasospasm	1 (2.2)
cCVSG at first spasmolysis (mean, range)	2.02 (0–3)
**Vasospasm-related characteristics**	
Early angiographic vasospasm (*n* = 26)	6
Occurrence of vasospasm (in days, median, range)	7.00 (0–19)
Vasospasm-related infarcts	35
BEHAVIOR, low risk (0–2)	2
BEHAVIOR, medium risk (3–6)	
BEHAVIOR, high risk (7–11)	6
On admission	34
Before intra-arterial spasmolysis	183
After 6 months (follow-up)	20
**Outcome after 6 months**	
mRS 0–1	14
mRS 2–3	6
mRS 4–5	19
mRS 6	6

TCD Transcranial Doppler, cCVSG clustered cerebral vasospasm grade, BEHAVIOR risk score for cerebral infarction including seven clinical characteristics.

**Table 3 jcm-11-02011-t003:** Agreement of visual classification and mathematical clustering.

		Agreement with Clustering in %					
cCVSG	*n*	NR 1	NR 2	NR 3	NR 4	NR 5	NR 6
0	45	88.9	82.2	73.3	62.2	75.6	77.8
1	41	68.3	31.7	19.5	12.2	24.4	29.3
2	97	89.7	73.2	93.8	94.8	90.7	84.5
3	51	74.5	43.1	51	51	58.8	60.8
Overall	234	82.5	61.1	67.5	64.5	69.2	68.4

Two-hundred-thirty-seven angiograms were graded by mathematical clustering and visual classification scheme. The agreement of each neuroradiologist (NR) with the clustering is listed according to each clustered cerebral vasospasm grade (cCVSG). Neuroradiologist 1 and 2 belong to group 1. For neuroradiologists 3 to 6 belonging to group 2, values of pM1 were added to the slides. The complementary criterion augmented the deviation in grades 0 and 1. The overall agreement of the second group did not exceed the first group without a complementary criterion.

## Data Availability

The analyzed datasets are provided at https://www.zenodo.org (26 February 2022) with the Digital Object Identifier Number: 10.5281/zenodo.5805604.

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
