# Peer review of "Standardized Classification of Cerebral Vasospasm after Subarachnoid Hemorrhage by Digital Subtraction Angiography"

_jcm, 2022, doi:10.3390/jcm11072011_

Round 1

Reviewer 1 Report

This study aims to introduce an easy-to-use standardized classification of CVS after SAH, with the aid of mathematical clustering. It is of importance as a clinical tool for the attending physicians and why not could be a new common metric system for the measurement of the vasospasm.

Only one point: ‘DSA was performed according to vascular regions with suspected CVS. Therefore, not all segments were necessarily examined in every patient at every time point, especially not the vertebrobasilar circulation.’  Did the authors perfume direct DSA comparisons between the constricted and the unaffected parts of the arterial tree of each patient? Or they based only on TCD measurements?  

Author Response

Response to Reviewer 1 Comments

Point 1: ‘DSA was performed according to vascular regions with suspected CVS. Therefore, not all segments were necessarily examined in every patient at every time point, especially not the vertebrobasilar circulation.’  Did the authors perform direct DSA comparisons between the constricted and the unaffected parts of the arterial tree of each patient? Or they based only on TCD measurements?

Response: Data are analyzed retrospectively. We did not perform a direct comparison of vessel diameters between the constricted and unaffected parts of the arterial segments of each patient. DSAs on admission were clustered as grades 0 – 2 (line 298 – 301). DSA on admission is suitable to assess anatomical variations, but vessels can be affected by early angiographic vasospasm up to 72 hours after SAH. Therefore, the DSA on admission was not used as baseline DSA for measurements.

Lines 113-120:

Monitoring included daily transcranial Doppler investigation (TCD) of cerebral arteries. If cerebral vasospasm was suspected based on altered level of consciousness or new neurological impairment (n=16, Tab. 1) or increased flow velocity in TCD above 120 cm/s in MCA or doubling within 24 hours (n=20). CT was performed in 44 patients to detect new territorial infarctions (n=5). CT perfusion analysis was additionally performed in cases with ambiguous findings (n=29). Six patients revealed a perfusion deficit leading to intra-arterial spasmolysis. Diagnostic DSA was performed if clinical worsening persisted, but TCD and CT-perfusion did not reveal CVS (n=1).

Lines 126 – 131:

Every patient who underwent intra-arterial spasmolysis obtained a diagnostic DSA run to assess the severity of angiographic cerebral vasospasm. DSA was performed according to vascular regions with suspected CVS. A visual comparison of the diagnostic DSA and DSA on admission, if applicable, was performed. Therefore, not all segments were necessarily examined in every patient at every time point, especially not the vertebrobasilar circulation.

The counterside was not always evaluated in diagnostic DSA. Angiographic vasospasm often affects both sides in different degrees of severity. Slight changes in vessel diameters might be underestimated visually. Unfortunately, reference values for threshold-based diagnosis of CVS in DSA are not established.

Reviewer 2 Report

The present article proposed visual classification scheme of CVS. Although several important limitations are present the study is well-written and designed. Of note, diagnostic DSA was performed only in case of persistent clinical worsening, with negative TCD and CT-perfusion investigations for CVS. Therefore, I suggest that the authors clearly state the potential generalization of their findings.

How come the two groups of neuroradiologists were uneven? 

You mention that ''The results of the second group deviated more from the mathematical clustering''. Although it can be graphically suspected, I suggest that statistical confirmation is also provided. To that end, please support every one of your observations with the relevant statistical confirmations. 

How come using a complementary diagnostic criterion led to worse reproducibility and greater deviation from the objective reference?

What is the clinical applicability of the proposed classification scheme (since higher grades were not related to poorer prognosis?).

Please remove lines 327-329.

Author Response

Response to Reviewer 2 Comments

Point 1: Of note, diagnostic DSA was performed only in case of persistent clinical worsening, with negative TCD and CT-perfusion investigations for CVS. Therefore, I suggest that the authors clearly state the potential generalization of their findings.

Response:

Data were complemented in lines 113-120:

Monitoring included daily transcranial Doppler investigation (TCD) of cerebral arteries. If cerebral vasospasm was suspected based on altered level of consciousness or new neurological impairment (n=16, Tab. 1) or increased flow velocity in TCD above 120 cm/s in MCA or doubling within 24 hours (n=20). CT was performed in 44 patients to detect new territorial infarctions (n=5). CT perfusion analysis was additionally performed in cases with ambiguous findings (n=29). Six patients revealed a perfusion deficit leading to intra-arterial spasmolysis. Diagnostic DSA was performed if clinical worsening persisted, but TCD and CT-perfusion did not reveal CVS (n=1).

Point 2: How come the two groups of neuroradiologists were uneven?

Response:

Raters of group 1 (n=2) pre-evaluated the classification scheme for improvements. Their data were not initially planned to be published. We expected a higher reproducibility after modification for the second group of raters (n=4).

Point 3: You mention that ''The results of the second group deviated more from the mathematical clustering''. Although it can be graphically suspected, I suggest that statistical confirmation is also provided. To that end, please support every one of your observations with the relevant statistical confirmations. 

Response: Thank you for this comment. We replaced Table 3. The agreement of mathematical clustering and grade-dependent ratings is shown for each neuroradiologist in detail.

Table 3. Agreement of visual classification and mathematical clustering

cCVSG

n

Agreement with clustering in %

NR 1

NR 2

NR 3

NR 4

NR 5

NR 6

0

45

88.9

82.2

73.3

62.2

75.6

77.8

1

41

68.3

31.7

19.5

12.2

24.4

29.3

2

97

89.7

73.2

93.8

94.8

90.7

84.5

3

51

74.5

43.1

51

51

58.8

60.8

Overall

234

82.5

61.1

67.5

64.5

69.2

68.4

Table 3. Two-hundred-thirty-seven angiograms have been graded by mathematical clustering and visual classification scheme. The agreement of each neuroradiologist (NR) with the clustering is listed according to each clustered cerebral vasospasm grade (cCVSG). Neuroradiologist 1 and 2 belong to group 1. For neuroradiologists 3 to 6 belonging to group 2, values of pM1 were added to the slides. The complementary criterion augmented the deviation in grades 0 and 1. The overall agreement of the second group did not exceed the first group without a complementary criterion.

Lines 276 – 296:

3.3. Rater-agreement and Reproducibility

The clustering distribution according to the grades 0-3 is shown in Figure 3 and Table 3. The results were re-evaluated after the grading by two interventional neuroradiologists (group 1: NR 1 and NR 2) with an agreement of 68.4%. There was no significant intragroup difference (p = .206), but the second rater had difficulties distinguishing CVSG 1 and 3. The highest agreement with the mathematical clustering was reached in CVSG 0 and 2.

For the second group, the classification was modified with an additional criterion to separate CVSG 2 from CVSG 3. The pM1 measured point less or equal to 1 mm with a 95% confidence interval was chosen as the threshold for CVSG 3. The measured values for pM1 were added on each slide. Four neuroradiologists (group 2: NR 3-6) used the modified classification. The results showed an increased reproducibility of 75.2 % with 100 % agreement of four raters. There were no significant intragroup differences (p = . 310).

CVSG 1 ratings decreased by half, whereas CVSG 2 ratings were significantly more distinct. The additional information of one vessel diameter remarkably influenced the raters of the second group in their decision. There was a highly significant intergroup difference (p <.001). The results of the second group deviated more from the mathematical clustering in cCVSG 0 and 1. Overall grades, the pM1 value augmented the interrater-reproducibility, but it did not remarkably influence the agreement for CVSG 3 ratings in group 2.

Point 4: How come using a complementary diagnostic criterion led to worse reproducibility and greater deviation from the objective reference?

Response: In Table 3, it is shown that the complementary criterion led to a shift from CVSG 1 to CVSG 2.

Added (lines 362 – 370):

Changes in pM1 diameters led to an overestimation of mild cerebral vasospasm. The visual classification scheme includes all proximal intradural vessels of the anterior circulation. Isolated changes of one segment are set in relation to the overall pattern without given values. The mathematical clustering takes account of all vessel diameters in the same manner. All raters of group 2 gave more weight to the pM1 diameter than the overall pattern of the angiogram. In severe vasospasm, narrowing of the intradural ICA is pivotal. The C7 diameter may better serve as a complementary criterion to define CVSG 3. Similarly, slight changes of C7 diameter may even lead to an overestimation of mild vasospasm not visually detectable.

Point 5: What is the clinical applicability of the proposed classification scheme (since higher grades were not related to poorer prognosis?).

Response:

Added (lines 435 – 441):

This classification can be used to develop a standardized escalation scheme for cerebral vasospasm. In CVSG 1, spasmolytic agents targeting smaller vessel segments, especially the anterior cerebral artery, are administrable. In contrast, mechanical spasmolysis for the M1 segment and the intradural ICA is more successful in CVSG 2 and 3. Pharmacological agents can be evaluated according to their vasodilating potential for different vessel segments. Moreover, the angiographic treatment success or nonsuccess can be evaluated standardized.

Point 6: Please remove lines 327-329.

Response: Done.

Round 2

Reviewer 2 Report

Thank you for considering my recommendations. This is a very interesting paper and will enhance existing literature.

Author Response

Thank you for your comments.